# Effect of Discharge Voltage on the Microstructure of Graphene/PEKK Composite Samples by Electromagnetic Powder Molding

**DOI:** 10.3390/polym15153256

**Published:** 2023-07-31

**Authors:** Fan Xu, Ming Gao, Hui-Xiong Wang, Xue-Lian Wu, Hong Liu, Chao Ma, Quan-Tong Yao, Hui-Yan Zhao

**Affiliations:** 1School of Mechanical Engineering, Jiangsu University, No. 301 Xuefu Road, Zhenjiang 212013, China; 2212003010@stmail.ujs.edu.cn (M.G.); whx8235@163.com (H.-X.W.); xlwu@ujs.edu.cn (X.-L.W.); 1000004949@ujs.edu.cn (H.L.); 2School of Mechanical Engineering & Automation, University of Science and Technology Liaoning, No. 189 Qianshan Centre Road, Anshan 114051, China; a18804258958@163.com (C.M.); tongtong282@126.com (Q.-T.Y.); 3School of Mechanical & Power Engineering, Yingkou Institute of Technology, No. 46 Bowen Road, Yingkou 115014, China; zhyyklg@163.com

**Keywords:** graphene/PEKK, microstructures, discharge voltage, compaction density, electromagnetic powder molding

## Abstract

The light weight, electrical conductivity, environmental friendliness, and high mechanical properties of graphene/PEKK composites make them popular in biomedical, electronic component and aerospace fields. However, the compaction density and carbonization of the specimen influence the microstructure and conductivity of the graphene/PEKK composite prepared by in situ polymerization, so electromagnetic-assisted molding was used to manufacture products to avoid carbonization and enhance the compaction density. The effects of different discharge voltages on the microstructure of the formed graphene/PEKK specimens were compared. Increasing the discharge voltage will lead to a closer distribution of flake graphene in the matrix to improve the compaction density, mechanical performance and conductivity. At the same time, the numerical analysis model was validated by comparison with the compaction density of the experimental results. Based on this research, the stress/strain distribution on the specimen was obtained with increasing discharge voltages.

## 1. Introduction

During the oil exploration and exploitation process, electromagnetic flowmeters (EMFs) have long been regarded as detection equipment in harsh underground environments [1]. According to the law of electromagnetic induction, when crude oil passes through the EMF, the magnetic field lines will be cut, and an electromotive force will be generated, which will then be detected by electrodes. The electrode materials of EMF are mainly composed of 316 L stainless steel, titanium, iridium and other metals [2]. However, semiconductor materials produced from polymers have received extensive attention compared with metallic materials. Researchers have demonstrated that adding a small amount of filler nanomaterials can significantly improve the mechanical properties and corrosion resistance of the polymer to meet the requirements of light weight [3,4,5]. At the same time, graphene has extremely high electrical conductivity and carrier mobility, reaching 2.0 × 10^5^ cm^2^·V^−1^·s^−1^, which exceeds those of existing semiconductor materials [6]. Graphene, as an efficient nanofiller material, can greatly change the properties of composites due to its ultrahigh aspect ratio and special planar hexagonal lattice structure [7]. In addition, graphene also possesses good thermal conductivity of up to 5000 W·m^−1^·K^−1^, which is 10 times higher than copper. Thus, graphene has become a preferred semiconductor-filling material [8,9].

Commonly, graphene is regarded as a filler nanomaterial, and some metal and nonmetal materials are regarded as matrices. PEKK is a base material that belongs to the same family as PEEK, and it has similar properties to PEEK but has less splash during in situ polymerization, less cost and higher safety. PEKK has a melting point of 380 °C and can work for a long time at temperatures below 250 °C. It is an important material for national defense and military industries because of its resistance to corrosion, wear and oil [10,11]. The graphene/PEKK composite material not only has the above characteristics but also high mechanical properties and electrical conductivity.

Previous studies compared graphene composites’ properties and formation methods with different matrices and reported the higher tensile strength of graphene/PEKK than other composites; moreover, Young’s modulus and conductivity can be improved by increasing the graphene content [12,13,14,15]. Differences in the performance of composite materials under different molding conditions were also investigated. The reinforcement effect of graphene is pronounced in composites. At present, increasing attention has been given to the formation and application of graphene composites, which have gradually become a research hotspot in the past decade [16]. The primary molding methods for graphene composites include additive manufacturing [17,18], hot pressing [19,20], spark plasma sintering [14], injection molding [21,22], extrusion molding [23], and electromagnetic pulse powder compaction [24], as shown in reference [25].

The advantage of additive manufacturing technologies such as melt deposition and laser sintering is that they can achieve near-net forming. Still, the printing speed is slow, and the requirements for material physical and chemical properties are high. At the same time, 3D products have significant anisotropy and high porosity, especially after adding reinforced materials—the anisotropy and porosity of the materials increase substantially after reinforcement [26]. Some companies have developed different series of products using injection molding. Carbon fiber and glass fiber reinforcement materials can significantly improve Young’s modulus and tensile strength of PEKK composite materials, with a tensile strength of over 250 MPa but lower conductivity [17].

The molding temperature of injection molding is higher than the material melting temperature, and the melting temperature of PEKK exceeds 360 °C, resulting in a longer cooling time and reduced production efficiency. At the same time, the internal stress of products formed by injection molding is relatively high, especially during mold conversion due to uneven cooling, resulting in product cracking. When processing products with complex shapes and larger sizes, product quality cannot be guaranteed; thus, the shape and size of the products are greatly limited [26]. The results further proved that the molding method is one of the main factors that affect the performance of composite materials.

The properties of graphene composites were not only improved through synthesis, but also different molding methods were used to improve the mechanical properties, electrical conductivity and thermal stability of graphene. This article uses electromagnetic-assisted forming methods to obtain samples under different process parameters. Based on the samples’ microstructure morphology, the process parameters’ influence on the microstructure is determined. At the same time, the influence of process parameters on mechanical and electrical properties is further defined. There are many studies on electromagnetic forming powder, but few studies on the powder forming of semiconductor materials, especially graphene/PEKK composites [24]. High-speed impact can overcome the defects of carbonization and excessive porosity caused by temperature and pressure by increasing compaction density at room temperature and is used to improve mechanical and electrical properties. This method has been applied to ceramic and metal powder forming, but there is little discussion on the high-speed impact forming of semiconductor materials. However, the cost is relatively high, and the equipment utilization rate is relatively low. This characteristic will limit the application of the electromagnetic forming method.

This research discusses the molding mechanism of graphene/PEKK composite powder from in situ polymerization under the electromagnetic-assisted graphene/PEKK composite molding process. Section 2 introduces the characteristics of graphene/PEKK composites and the electromagnetic-assisted powder molding equipment. Section 3 mainly discusses the microstructure and morphology of the electromagnetic-assisted graphene electrode rod under different discharge parameters. All simulations were carried out using the dynamic code ABAQUS/Explicit, according to Fan et al. [27]. ABAQUS/Explicit is a display solver; it supplies a subroutine interface UMAT of user-defined material constitutive models, allowing users to use Fortran or C++ to write the source code, and then the constitutive model is embedded into ABAQUS. Experimental and numerical analyses are combined to further determine the relationship between the compaction density, microstructure and conductivity of the specimens.

## 2. Experiment and Method

### 2.1. Material Characterization

PEKK is a high-performance thermoplastic that belongs to the polyaryletherketone family [14,15,24]. It has high thermal stability, chemical resistance and mechanical properties. Its high glass transition temperature (*T_g_* = 140 °C) and melting point (*T_m_* = 320 °C) make it suitable for aeronautic applications [27,28]. In the present work, reduced graphene was purchased from Shenzhen Yuewang Energy Saving Technology Service Co., Ltd. (Shenzhen, China). A previous study [14,15,24] reported the preparation of graphene/PEKK composite powders and related material characterization techniques. The thermal properties of the self-prepared graphene/PEKK composite powder were characterized by thermogravimetric analysis (TGA) and differential scanning calorimetry (DSC) before molding the conductive graphene/PEKK composite. TGA (TGA-NETZSCH STA 449F3, Selbu, Germany) was conducted from 50 °C to 1000 °C at a ramp rate of 10 °C/min. DSC (DSC-NETZSCH STA 449F3, Selbu, Germany) was carried out between 0 °C and 450 °C at a heating/cooling rate of 10 °C/min [15].

The preparation methods and related characterization techniques are the same for graphene/PEKK composite powders with different graphene contents. Reference [15] show the particle morphology of the PEKK powder and graphene/PEKK composite powder under a stereo microscope. Figure 1a,b shows scanning electron microscopy (SEM) images of PEKK and the graphene/PEKK composite. Based on the comparison of PEKK and the graphene/PEKK composite, the flake graphene is distributed in the PEKK matrix. These flakes will ensure the conductivity of the graphene/PEKK composite sheets.

Laboratory-prepared PEKK was used as a matrix for polymerization with 4.5% graphene. The effect of graphene on the thermal stability of PEKK composites was investigated by DSC and TGA. As shown in Figure 2a, graphene reduces the melting point of the PEKK composite material; although the thermal weight loss rate is accelerated, it still has good thermal stability. Graphene starts to decompose at 530 °C and only decomposes by 48% when it reaches 1000 °C. Hence, graphene can be used to synthesize graphene/PEKK composites that can work under harsh environmental conditions at high temperatures (Figure 2b). Therefore, the high-speed cold molding method is suitable for composite powder formation.

### 2.2. Electromagnetic-Assisted Molding Device

The ARCHIMEDES VEMP-80 electromagnetic molding machine developed by Archimedes Industrial Technology Co., Ltd., Beijing, China, was used (Figure 3). Figure 3a shows the electromagnetic-assisted hydro-forming device, Figure 3b shows the mold assembly, and Figure 3c shows the geometry of the tools (diameter = 10 mm). The graphene/PEKK powder is placed inside the mold and then set at different charging voltages using the control cabinet to charge. When the voltage across the capacitor bank reaches the set value, the charging switch K1 is turned off to complete the charging process of the capacitor. At this time, switch K2 is closed, and the high voltage at both ends of the capacitor group discharges the flat coil instantaneously. When the current flows through the flat coil, an induced magnetic field is formed near the coil.

In Figure 3b, a drive sheet placed below the flat coil generates eddy currents under an induced magnetic field and forms a closed-loop circuit. Electromagnetic induction forms an induced magnetic field near the drive sheet, and the interaction of the two fields generates a strong pulse force on the plate. The drive sheet is placed above the press and in contact with the flat coil. The flat coil is connected to the capacitor by a wire and is held in place by the hydraulic device that presses the molding device. The drive sheet drives the punch to compact the graphene/PEKK powder. The device mainly uses the hydraulic system to control the return stroke of the punch to achieve a working cycle within a few milliseconds. The working coil is a flat spiral coil wound with a cross-sectional area of 2 mm × 20 mm purple copper wire. The key component is the drive sheet, which generates an induced magnetic field during discharging, in contrast to the magnetic field generated by the coil.

## 3. Results and Discussion

### 3.1. Discharge Current and Electromagnetic Force

An Agitek coil without an integrator is used for measurement during the electromagnetic molding process. The input current is proportional to the output voltage signal, the output sensitivity is 6.8 mV/kA, the operating frequency is 50 Hz, and the relationship between the input current and the output voltage is obtained as follows:(1)i(t)=1000×1000t(t)/6.8×50=2941u(t)

Discharge currents were measured at discharge voltages of 7, 8 and 9 kV (Figure 4a). The first peak of the current at different discharge voltages is approximately 0, mainly due to the influence of the operating frequency of the Roche coil without the integrator, which causes the measured sine wave phase to be more advanced than the actual value. The phase is 90° ahead when the operating frequency is 50 Hz, the peak value of the discharge current increases gradually with the discharge voltage, and the discharge period is basically unchanged. In fact, it also shows that powder particles complete deformation under the action of inertial force (2 μm).

A pressure sensor is used to measure the electromagnetic pulse pressure. The sensor contains strain gauges, which are encapsulated in the sensor with PVDF adhesive. When the pressure sensor is subjected to an impact force, it is deformed, and an electrical charge is generated on the surface. The charge is transmitted via wires to a charge amplifier for amplification. The charge signal is collected by a data acquisition instrument and transmitted to a computer for display. The pressure sensor (Archimedes Industrial Technology Co., Ltd.) has a measurement range of 900 kN and a sensitivity of 0.036 pC/N. The test uses a charge amplifier with a model number VK102 (Shenzhen Micro Precision Electronics Co., Ltd., Shenzhen, China), a charge input range of 0 to ±5000 pC and a sensitivity of 100 pC/100 mV. The relationship between the impact force and the voltage signal is expressed as
(2)F=U/0.036
where *F* is the impact force in N, and *U* is the output voltage in mV.

Discharge voltages of 7 kV, 8 kV and 9 kV were used, and their periods were basically the same as those of the measured discharge currents. As the discharge voltage rises, the impact force gradually increases, and the increase in the first peak is the most obvious. The pressure sensor can only measure the pressure but not the tension, so the negative half-axis of the force is zero. In fact, the energy utilization rate of electromagnetic molding is low, generally at 10% to 20%, and can hardly exceed 30%. The stiffness of the frame and die under high-speed impact directly affects the electromagnetic force acting on the powder. The impact load measured on the frame is shown in Figure 4b.

### 3.2. Microstructure Characteristics

The equipment and mold were used for the graphene/PEKK composite powder, and then the discharge voltage was adjusted to obtain graphene/composite tablets (4–12 mm) with different compaction densities. This experiment was designed with three different discharge parameters for electromagnetic-assisted powder compaction monitored during the molding process. The effect of temperature rise changes generated by the high-speed impact on the compaction density and microstructure was ignored. Compaction density is one of the main factors affecting mechanical, physical and chemical properties and directly affects the microstructure.

Compaction density is an important parameter to measure the powder forming quality. The higher the compaction density is, the higher the mechanical properties and electrical conductivity. Figure 5 shows the compaction density variations in graphene/PEKK composite electrode rods at different discharge voltages of 7 kV, 8 kV and 9 kV. The compaction density increases with increasing discharge voltage/electromagnetic force for a 10 mm diameter rod. The compression force per unit area is the main factor that affects the compaction density. The contact between the particles changes from elastic to plastic in the mold. The deformation rate increases slowly after the plastic deformation reaches a certain level, consistent with the particle plastic deformation law. The macroscopic physical phenomenon reveals the microstructure evolution law.

Ten equal samples of 0.8 g of graphene/PEKK composite powder were weighed in the laboratory and subjected to different discharge voltages to obtain graphene/PEKK electrode rods of different heights. The micromorphology of the upper and lower surfaces in contact with the upper and lower molds is shown in Figure 6. Figure 6a shows the micromorphology near the surface of the punch, and Figure 6b shows the micromorphology near the end of the lower die. There are fewer micro holes on the surface near the punch side and more on the surface near the lower die side. The graphene flake is evenly distributed on the surface, thus ensuring the conductivity of the specimen surface. The electrode rods obtained by the different discharge voltages were quenched, and the SEM microstructures of the sections quenched along the diameter were obtained at 7, 8 and 9 kV at different magnifications (Figure 7). In contrast to pure PEKK, flake graphene is distributed in the PEKK matrix as the filler material of the composite. The inertial force generated by the electromagnetic force acts as a macroscopic force to deform the particles simultaneously in the radial and axial directions, eventually molding a structurally stable electrode rod in the mold.

The flake graphene is distributed in the direction perpendicular to the electromagnetic force. The compaction density increases with increasing discharge voltage/electromagnetic force; the tighter the flake structure, the stronger the directional distribution. Thus, the influence of temperature on the molding quality and microstructure during molding can be ignored, and molding is assumed to be a cold molding process with high-speed impact graphene/PEKK composite powders having electrical conductivity. Still, the movement of powder particles is more violent under the effect of high-speed impact and an induced magnetic field. The kinetic energy of the particles increases under high-speed impact to form a strong bond. With increasing inertial force, the carrier mobility of the flake graphene structure is large, the electrical conductivity is stronger, and the structure is more stable.

### 3.3. Conductivity and Mechanical Performance

Conductivity is the basic property of graphene/PEKK composite powder and is the basis for judging a reasonable molding method and process. Figure 8 shows the electrical conductivity of the graphene/PEKK electrode rod at different discharge voltages. As the discharge voltage was increased, the electrical conductivity of the electrode rod also increased. The electrical conductivity at discharge voltages of 8 kV and 9 kV was greater than that at 7 kV.

The average electrical conductivity of the graphene/PEKK electrode rod increased from 1.46 S/m at 8 kV to 1.72 S/m at 9 kV. When the discharge voltage was 9 kV, the maximum value of electrical conductivity at different points of the electrode rod was 2.65 S/m, and the areas with relatively high density had high electrical conductivity. The infrared thermal imager found no obvious increase in the temperature on the surfaces of the tools during molding. Therefore, high-temperature sintering cannot occur during the molding process. The electrode rod must be post-treated to improve its density if the electrical conductivity is to be improved further. In fact, the factor related to the conductivity is mainly the distribution of graphene flakes.

The application of graphene/PEKK in engineering is still inseparable from its mechanical properties. The discharge voltage shown in the previous test is the main factor affecting the compaction density, which affects not only the electrical conductivity but also the mechanical properties. Figure 9 shows the analysis of the mechanical properties, including the surface hardness, of the electrode rod at different discharge voltages. The graphene/PEKK composite powder had a mass of 0.4 g and relative densities of 0.899, 0.960, and 0.962. The hardness distribution on the electrode rod reflected the relative density of the region. With the increase in the discharge voltage, the relative density of the electrode rod increased, and the surface hardness of the rods subsequently also increased. Therefore, increasing the discharge voltage can effectively improve the hardness of the electrode rod. This effect was basically consistent with the effect of the discharge voltage on the relative density of the electrode rod. Generally, to increase the adhesion between particles, post-curing treatment is carried out to improve the compaction density and the mechanical properties of specimens.

### 3.4. Numerical Analysis

(1)FE Analysis

During the process of electromagnetic forming of graphene/PEKK composite powder, it is difficult to monitor the particle deformation and adhesion under the action of stress waves, so it is necessary to establish a suitable finite element numerical analysis model. The tool diameter was 10 mm, and the other parameters are shown in Figure 3c. The mass of the graphene/PEKK composite powder was 0.82 g, and the initial height was 21 mm. Given that the tool material was hard alloy steel with hardness exceeding 70 HRC after quenching, the finite element model of the tool can be regarded as a rigid body. The model was simplified, as shown in Figure 10. The powder unit type was C3D8I, the unit size was 1 mm, and the number of divided units was 2604. This study found that mesh size had little effect on simulation accuracy but greatly affected operation speed. When the grid size was set to 0.5 mm, the number of grids reached 19,656, and the operation time was 6 h, which was far longer than the operation time of 5 h when the grid size was 1 mm. The grid size was 1 mm and was considered the cell size. At the same time, some special points with red circle were selected to investigate the distribution of stress, strain and density in electrode rod. 

The driving sheet transmitted the stress wave of the electromagnetic pulse force to the punch as an external load for the electromagnetic formation of the graphene/PEKK composite powder. The contact algorithm adopted the penalty function method, the friction model adopted the Coulomb friction model, and the friction coefficient was 0.1 [25]. These red points (Figure 10) are located in the shear bang of the bar, so they were selected to analyze the stress/strain and velocity distribution on the bar.

(2)Model validation

This study quoted the Doraivlue model on PEKK revised by Fan et al. [27,29] (Equation (3): *m* = 0.08, *n* = 4). To determine the accuracy of the model, test samples with discharge voltages of 7 kV, 8 kV and 9 kV are taken as test samples, and the prediction accuracy of the model is verified with the compaction density criterion. With the increase in the discharge voltage, the relative density of the electrode rod also increased, and the density distribution on the electrode rod became increasingly uniform. However, the difference between the highest and the lowest relative densities continued to exceed 0.1 mainly because the initial height:diameter ratio exceeded 2 and the energy loss during the pressing process was large, resulting in the low relative density at the bottom of the electrode rod. SDV2 expresses the relative density distribution (Figure 11). As the discharge voltage was increased, the compaction density also increased.
(3)$$(1.86\rho +1.14)J_{2}^{\prime }+m(1-\rho )J_{1}{}^{2}-\frac{\rho^{n}-\rho_{c}{}^{n}}{1-\rho_{c}{}^{n}}\sigma_{s}{}^{2}=0$$
(4)$$(1.86\rho +1.14)J_{2}^{\prime }+0.08(1-\rho )J_{1}{}^{2}-\frac{\rho^{4}-\rho_{c}{}^{4}}{1-\rho_{c}{}^{4}}\sigma_{s}{}^{2}=0$$

Figure 12 shows the relative densities at discharge voltages of 7, 8, and 9 kV obtained by comparing the experimental results with the prediction results of the modified Doraivelu model. The relative density increased as the discharge voltage was increased. Specifically, when the discharge voltages were 7, 8, and 9 kV, the experimental relative densities were 0.743, 0.916, and 0.957, respectively, and the predicted relative densities were 0.729, 0.895, and 0.95, respectively. These results showed that the error between the simulation and experimental results was small, indicating that the modified Doraivelu model [30] can effectively predict the relative density at different discharge voltages. The prediction ability of the Doraivelu model of graphene/PEKK composite powder was determined, and the distribution laws of compaction density, stress–strain and deformation rate during the molding process were predicted by this model.

(3)Stress/strain distribution on the electrode rod

Given that the electromagnetic pulse compaction model was axisymmetric, its 1/4 model was selected. The graphene/PEKK powder experienced a complex flow process because the electromagnetically assisted molding is a high-speed molding process. The stress, strain, and relative density distribution on the electrode rod, which are important factors affecting forming quality and geometry, cannot be monitored by sensors. Figure 13 shows the stress distribution. When the discharge voltages were 7, 8, and 9 kV, the maximum stresses were 36.804, 82.731, and 124.32 MPa, respectively. Therefore, as the discharge voltage was increased, the maximum stress distributed on the electrode rod also increased.

Similar to the stress distribution (Figure 13), the maximum strain increased as the discharge voltage increased, as shown in Figure 14. During high-speed impact, the force between particles and between particles and the tools caused the strain to concentrate on the nearby tools. Relative compaction density is an important indicator for measuring product quality (Figure 12).

The stress, strain, and relative density at eight different distribution positions on the electrode rod were described in accordance with the points shown in Figure 10. When the time was close to 2 ms, the stress reached its highest value, and the strain and relative density also changed. The stress, strain, and relative density changed with the radial radius. Although nodes 9 and 13 were distributed along the same longitude, node 9 was near the punch position (high latitude), and node 13 was near the gasket (low latitude). Therefore, the stress at node 9 decreased rapidly after yielding during high-speed impact. At the same time, the strain and relative density at each node changed with time, whereas each node’s stress, strain, and relative density were almost the same before 2 ms. Given that the electromagnetically assisted molding of graphene/PEKK powder is a high-speed impact process, the strain rate between particles was close such that particle deformation was uniform and fracture failure caused by stress concentration did not occur easily. However, considering that controlling the discharge voltage or discharge energy remains important for controlling quality, the relative density should be increased as much as possible.

When the duration of electromagnetically assisted powder molding exceeded 2 ms, the relative density of the electrode rod was further improved mainly by the plastic deformation of the particles. The impact force from the punch was attenuated, and the input energy was reduced. Part of the energy was consumed due to friction between particles, resulting in a relatively obvious density gradient. The density was large and gradually decreased toward the central area of the axis. The evolution of stress/strain/density at different points of the bar can be described as shown in Figure 15.

The electromagnetically assisted powder molding process has high-speed loading and energy. The inertial force causes particles to collide incompletely and inelastically before 2 ms. The time–velocity curves of the punch at different discharge voltages are shown in Figure 16. During the molding process, the impact speed first increased and then decreased. During the first peak time, the velocity showed the fastest increase, and the impact velocity further increased under the action of inertial force, whereas acceleration gradually decreased. The impact velocity increased continuously as the discharge voltage was further increased and reached 4.15 m/s when the discharge voltage was 7 kV.

## 4. Conclusions

Graphene/PEKK composite powder was molded through electromagnetically assisted pulse compaction, and graphene/PEKK composite electrode rods were obtained under different discharge voltages.

(1)Electromagnetically assisted powder molding equipment was used to process graphene/PEKK composite electrode rods at different discharge voltages, such as 7 kV, 8 kV, and 9 kV. The discharge voltage was adjusted to improve the compaction density during the powder molding process and to enhance the electrical conductivity and mechanical properties of the electrode rods.(2)Under high-speed impact, graphene is uniformly dispersed in the PEKK matrix and along the direction perpendicular to the impact force. The flake graphene structure is relatively compact with increasing discharge voltage (electromagnetic force). The relative density values are 0.957 at 9 kV, 0.916 at 8 kV and 0.734 at 7 kV. Thus, the compaction density increases as the discharge voltage increases, and the flake graphene structure is tight, resulting in an increase in the carrier motion rate.(3)During the electromagnetically assisted molding of the graphene/PEKK composite powder, the electrical conductivity and hardness increased with increasing discharge voltage, indicating that the compaction density affected the mechanical and electrical properties of the graphene/PEKK composite. When the discharge voltage reached 9 kV, the conductivity of the electrode rod reached 2.65 S/m on the surface.

The molding of large components is currently limited, and the most important factor is the capacitance of electromagnetic molding equipment. The utilization rate of electromagnetic forming equipment is currently low, usually at 15%. Therefore, in future research, we will not only explore the molding process but also carry out engineering research to improve the utilization rate of electromagnetic forming equipment.

## Figures and Tables

**Figure 1 polymers-15-03256-f001:**
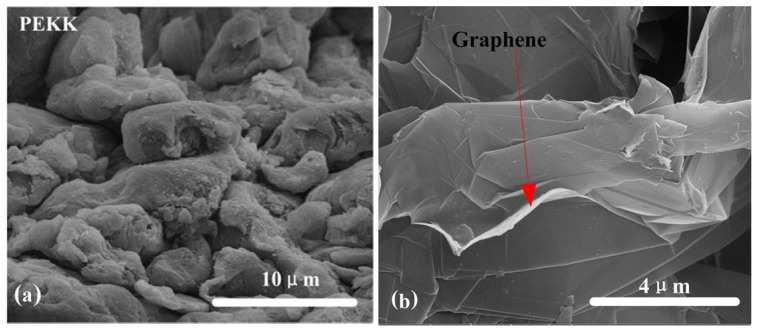
Microanalysis of PEKK and graphene/PEKK composite: (**a**) PEKK sheet; (**b**) graphene/PEKK sheet.

**Figure 2 polymers-15-03256-f002:**
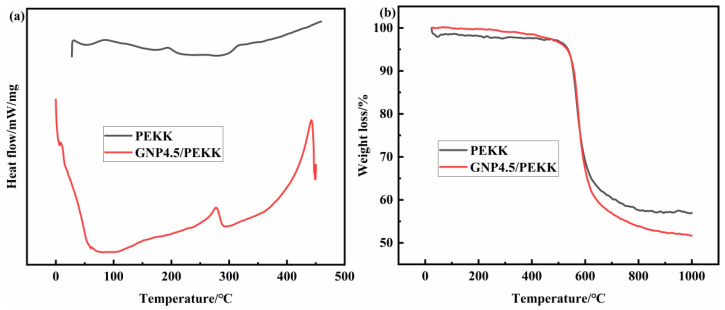
Thermal performance test chart of PEKK and graphene/PEKK: (**a**) DSC, (**b**) TGA.

**Figure 3 polymers-15-03256-f003:**
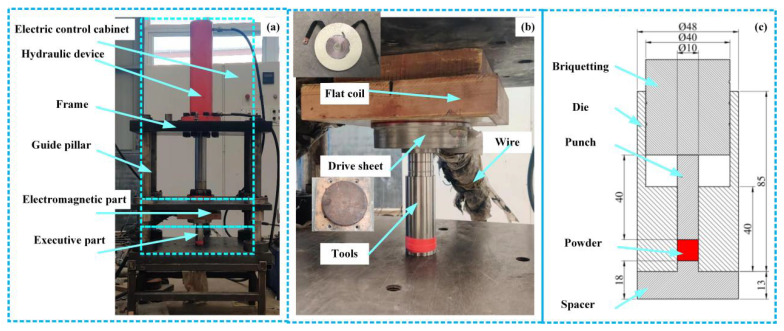
Experimental device: (**a**) frame of the electromagnetic device; (**b**) pair of tools; (**c**) geometrical size of the tools.

**Figure 4 polymers-15-03256-f004:**
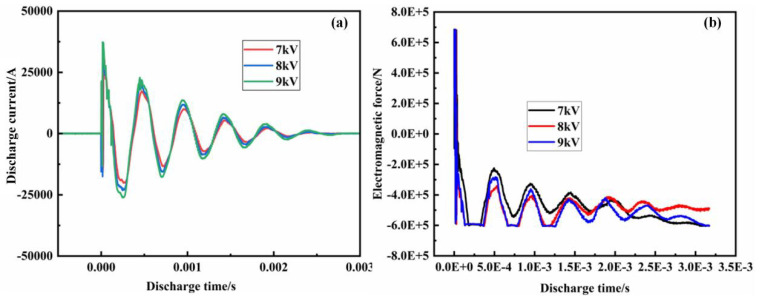
The discharge parameters: (**a**) discharge current; (**b**) electromagnetic force.

**Figure 5 polymers-15-03256-f005:**
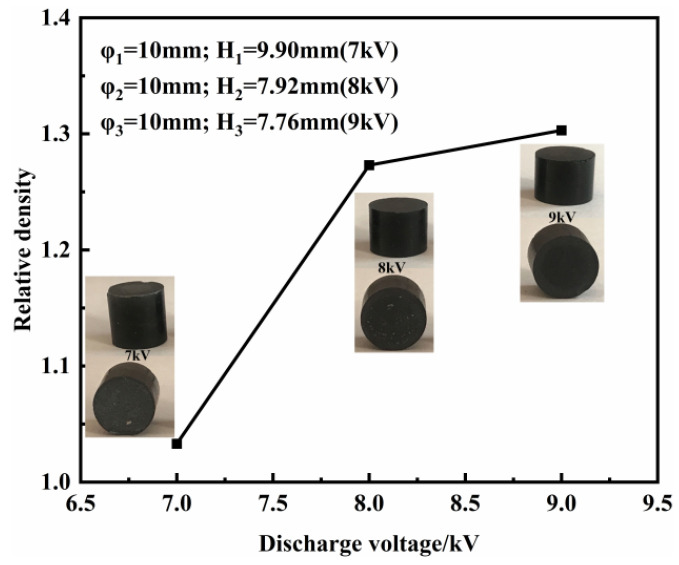
Compaction density under different discharge voltages.

**Figure 6 polymers-15-03256-f006:**
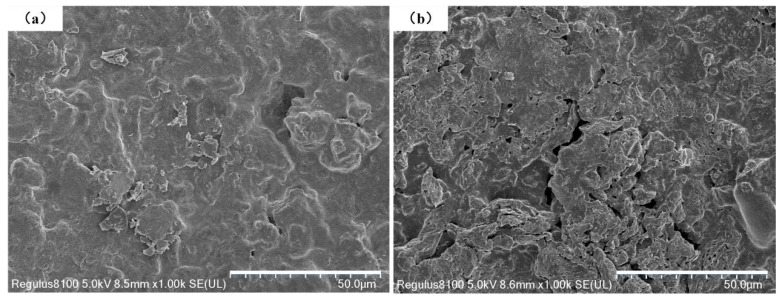
SEM analysis for the section (**a**) near the die; (**b**) near the punch.

**Figure 7 polymers-15-03256-f007:**
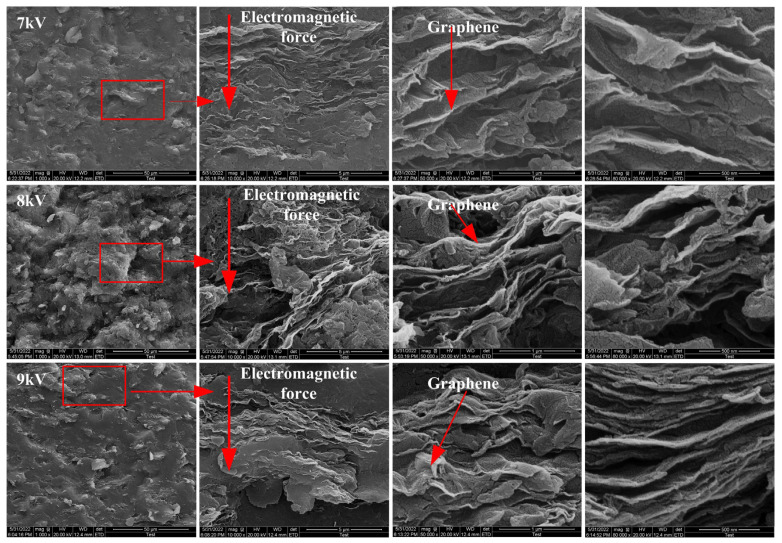
Microstructure of the graphene/PEKK composite materials under different discharge voltages.

**Figure 8 polymers-15-03256-f008:**
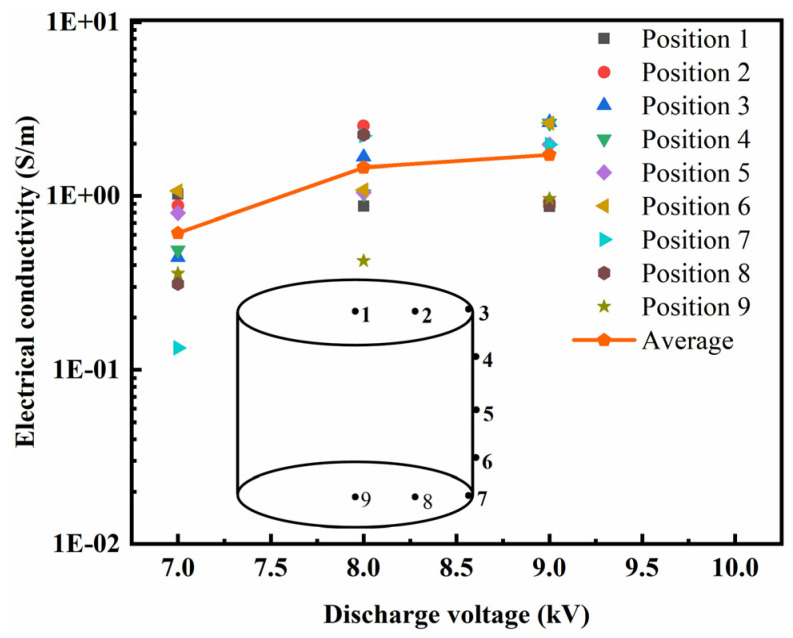
The conductivity of different points under different discharge voltages.

**Figure 9 polymers-15-03256-f009:**
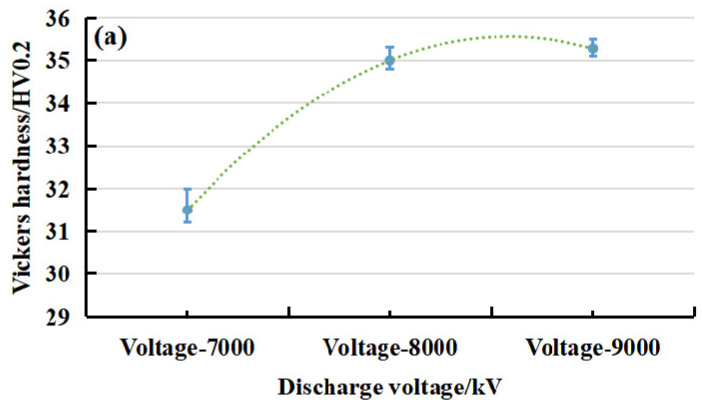
Mechanical performance of the electrode rod: (**a**) The hardness of different points under different discharge voltages; (**b**) The compressive strength of the electrode rod under different discharge voltages.

**Figure 10 polymers-15-03256-f010:**
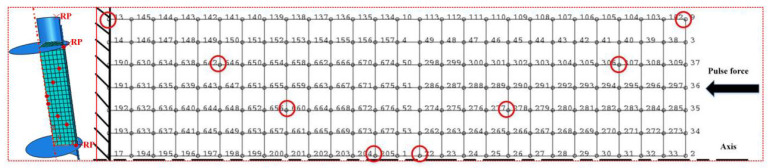
Finite element model of powder compaction.

**Figure 11 polymers-15-03256-f011:**
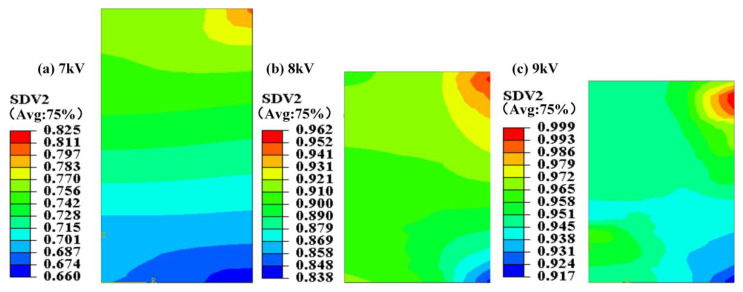
The density distribution on the electrode rod under different discharge voltages.

**Figure 12 polymers-15-03256-f012:**
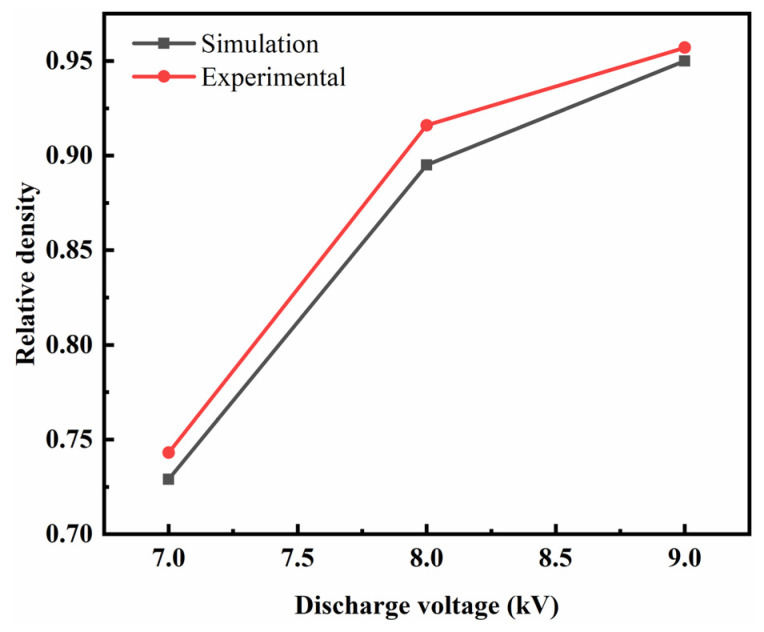
The experimental results are compared with the numerical results of the compaction density.

**Figure 13 polymers-15-03256-f013:**
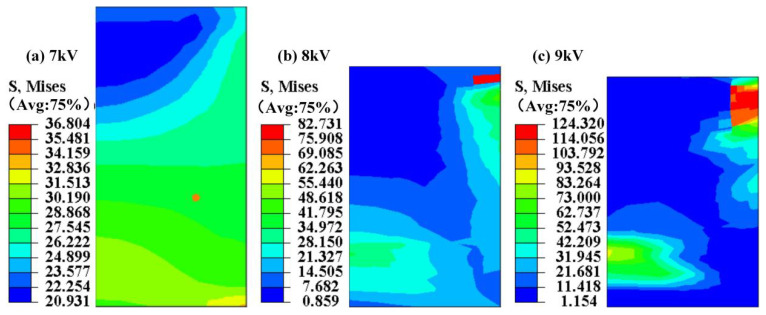
The stress distribution on the electrode rod under different discharge voltages.

**Figure 14 polymers-15-03256-f014:**
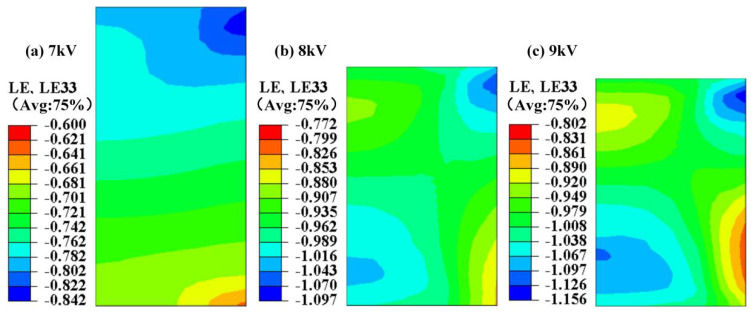
The strain distribution on the electrode rod under different discharge voltages.

**Figure 15 polymers-15-03256-f015:**
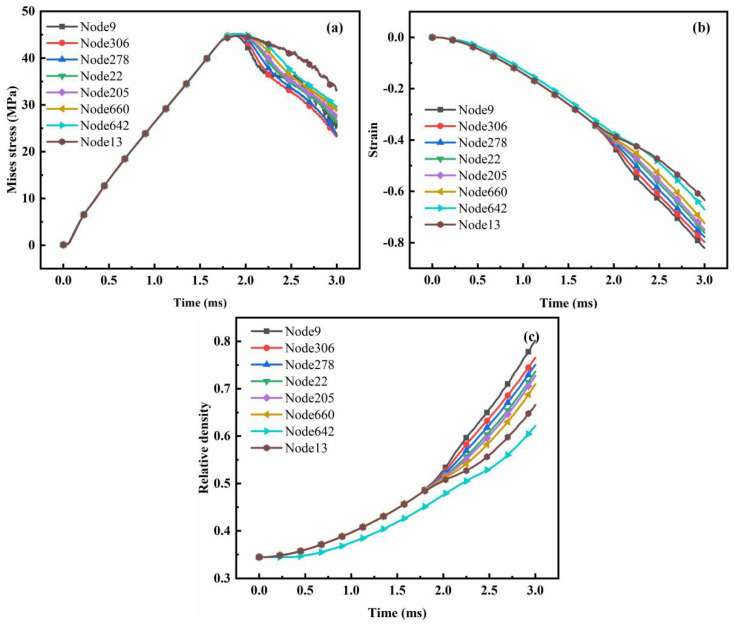
Evolution of stress/strain/relative density at different points of the electrode rod under the different technique parameters: (**a**) Evolution of stress; (**b**) Evolution of strain; (**c**) Evolution of relative density (7 kV).

**Figure 16 polymers-15-03256-f016:**
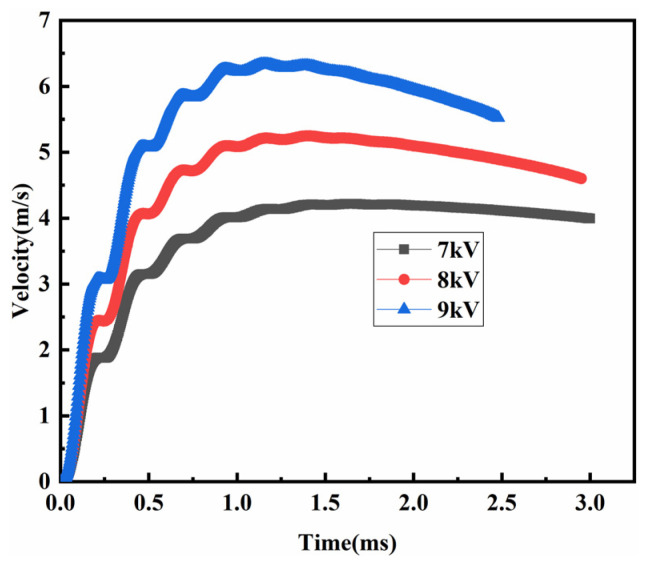
The evolution of the velocity under electromagnetic processing.

## Data Availability

Data sharing is not applicable to this article.

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
