# Peer review of "Effect of Discharge Voltage on the Microstructure of Graphene/PEKK Composite Samples by Electromagnetic Powder Molding"

_polymers, 2023, doi:10.3390/polym15153256_

Round 1

Reviewer 1 Report

In the present study, the authors demonstrated a novel method for fabricating graphene/PEKK composites by electromagnetic-assisted molding. They found that increasing the discharge voltage leads to a tighter distribution of graphene flakes in the matrix, resulting in improved compacted density, mechanical performance and conductivity of the product. This study is well-designed and logically based on the results, so it deserves to be published. Only the quality of the IR spectra is very disappointing. To clearly understand the information from the IR studies, the authors need to depict the spectra in the wavenumber range of 1850-600 cm–1.

Considering the above-mentioned point, I suggest this manuscript for publication after minor revision.

Author Response

Dear Editor:

We would like to thank Polymers for giving an opportunity to resubmit our manuscript (Effect of Discharge Voltage on the Microstructure of Graphene/PEKK Composite Samples By Electromagnetic Powder Molding (Polymers-2505563)). We thank the reviewers for their careful reading and thoughtful comments on the previous draft. We have carefully taken their comments into consideration in preparing our manuscript, which has resulted in a paper that is clearer, more compelling and broader. We appreciate the constructive criticism and suggestion. We addressed all the points raised by the reviewers, as summarized below.

Reviewer 2 Report

The manuscript by Xu et al. entitled “The effect of discharge voltage on the Microstructure of Graphene/PEKK Composite Samples By the Electromagnetic Powder Molding” presents an interesting work Graphene/PEEK polymer composites. The manuscript is well written and easy to follow. Overall, this research contributes to the understanding and optimization of the manufacturing process for graphene/PEKK composites, enabling enhanced material properties for various applications. However, some minor corrections should be applied before final publication.

1.      More recent references should be cited in introduction to justify the originality of this work.

2.      The graphene flakes are not visible in Fig. 1. The authors should use supporting experiments to prove the existence of graphene.

3.      It is better to label the peaks in Figure 2 at specific wavenumber or wavelength.

4.      What is special about Fig. 6? It can be deleted.

5.      How did the authors measure electrical conductivity in Fig. 10?

6.      Although the effects of various discharge voltages on the microstructure of graphene/PEKK specimens were contrasted, other significant properties like thermal stability, electrical conductivity, and long-term durability were not thoroughly examined in the study. These additional studies would give us a more thorough understanding of the performance of the composite.

7.      The accuracy of the numerical analysis model may not be sufficiently supported by its validation based solely on compacted density. It would be advantageous to validate the model by comparing additional significant mechanical or electrical properties or by using a wider range of experimental data.

8.      For large-scale manufacturing, it is crucial to take into account the practicality and scalability of the electromagnetic-assisted molding technique. The application of the research should be improved by further discussion of the difficulties, restrictions, and viability of applying this method in industrial settings.

Minor polishing required. 

Author Response

(The authors gave the same response as above.)
